# Optimization of Germination Conditions for Enriched γ-Aminobutyric Acid and Phenolic Compounds of Foxtail Millet Sprouts by Response Surface Methodology

**DOI:** 10.3390/foods13203340

**Published:** 2024-10-21

**Authors:** Shibin Yu, Chunqiu Li, Xiaoyan Wang, Daniela D. Herrera-Balandrano, Joel B. Johnson, Jinle Xiang

**Affiliations:** 1Faculty of Food & Bioengineering, Henan University of Science & Technology, Luoyang 471023, China; 18837649619@163.com (S.Y.); lichunqiu1999@163.com (C.L.); wangtuomasi430@gmail.com (X.W.); 2Henan International Joint Laboratory of Food Green Processing and Safety Control, Henan University of Science & Technology, Luoyang 471023, China; 3School of Life Sciences, Nantong University, Nantong 226019, China; daniela.herrera@ntu.edu.cn; 4Centre for Nutrition and Food Sciences, Queensland Alliance for Agriculture and Food Innovation (QAAFI), The University of Queensland, Brisbane, QLD 4072, Australia; joel.johnson@cqumail.com

**Keywords:** foxtail millet, germination conditions, GABA, polyphenols, response surface method

## Abstract

The optimum germination conditions for foxtail millet sprouts enriched with γ-aminobutyric acid (GABA) and antioxidant polyphenols were investigated. From single-factor experimental results, both the GABA level and total phenolic content (TPC) were more significantly affected by soaking temperature and time, and concentration of sucrose culture solution. Response surface methodology (RSE) was used to optimize the germination conditions of foxtail millet sprouts, where the interaction between soaking temperature and sucrose concentration exhibited a significant (*p* < 0.05) effect on TPC, and the interaction between soaking time and sucrose concentration displayed a significant (*p* < 0.05) effect on GABA content. The optimal germination conditions for TPC and GABA enrichment of foxtail millet sprouts were soaking at 31 °C for 4.5 h and germinating at 35 °C with 4.5 g/L sucrose solution for 5 days. Under the optimized conditions, the TPC and GABA content of foxtail millet sprouts were 926.53 milligrams of ferulic acid equivalents per 100 g dry weight (mg FAE/100 g DW) and 259.13 mg/kg, separately, with less difference from the predicted values of 929.44 mg FAE/100 g DW and 263.60 mg/kg, respectively. Collectively, all the individual phenolic compounds increased significantly (*p* < 0.05) by optimization, except for *cis*-*p*-coumaric acid and *cis*-ferulic acid in bound. The results provide a practical technology for suitable germination conditions to improve the health components of foxtail millet sprouts and increase their added value.

## 1. Introduction

Foxtail millet (*Setaria italica* (L.) P. Beauv.), a member of the *Gramineae* family, is one of the world’s oldest crops and the sixth most important cereal grain [1]. Foxtail millet is drought- and cold-resistant and cultivated mainly in arid and semi-arid areas, with Asia and Africa being the main growing areas. Foxtail millet contains a variety of essential nutrients, such as minerals, vitamins, carbohydrates, dietary fiber, and proteins [2]. Although the dietary fiber in millet cannot be digested and absorbed by the human body, it can promote intestinal peristalsis and prevent digestive tract diseases. The proteins can maintain the constant osmotic pressure of plasma colloids, have the function of transporting plasma human albumin, and enhance the body’s resistance. Essential minerals maintain the health of muscles and bones, and vitamins, especially abundant vitamin B in foxtail millet, can support energy metabolism, brain function, and healthy skin [3]. In addition to micronutrients and macronutrients, it is a significant source of phytochemicals, particularly phenolic compounds, carotenoids, and γ-aminobutyric acid (GABA), which have demonstrated efficacy in mitigating the risk of chronic diseases, including hypertension, dyslipidemia, and hypercholesterolemia [4]. Millet can be processed into various food products because of its good nutritional value, such as millet-based beverages, bread, porridge, flatbreads, pasta, cereal energy bars, and gluten-free snacks, and it has the potential to help solving global food security problems [3]. Phenolic compounds are essential bioactive ingredients in cereal grains, with a wide range of health benefits such as antioxidant, anti-inflammatory, and anti-allergic functions. GABA, an important non-protein amino acid, is a kind of inhibitory neurotransmitter which is involved in a variety of physiological activities in human body, with unique physiological properties including calming and tranquilizing, anti-stress and brain health, lowering blood pressure, anti-anxiety, and so on [5].

Germination is an effective technique for enhancing the nutritional content and overall quality of grains for human consumption. Extremely complex physiological and biochemical reactions occur within grains during germination and are accompanied by the interconversion of endogenous components and production of new compounds, including changes in nutrients and especially bioactive phytochemicals, such as phenolics and GABA [6]. It has been reported that germination promotes the synthesis of polyphenols and GABA in some grains [5,7]. It has been reported that GABA contents of millet were increased by germination [8]. Throughout germination, some polyphenol synthases are expected to be active, with their activities progressively increasing in tandem with the germination process, leading to notable alterations in phenolic patterns and antioxidant activities [6]. Additionally, the activities of glutamic acid decarboxylase (GAD) and protease elevate during sprouting, resulting in a substantial buildup of GABA [9]; thus, the optimal germination conditions could significantly influence the contents of GABA and polyphenols in grains. Sharma et al. applied a central composite rotatable design to optimize the germination conditions of millet, which resulted in a 5.07-fold increase in GABA [10]. Bai et al. optimized the germination conditions for increasing GABA content of germinated millet using a Box–Behnken design and achieved 42.9 mg/100 g of GABA under their optimal conditions [11]. Sharma et al. used a central composite rotatable design (CCRD) in response surface methodology to optimize the levels of total phenolics and total flavonoids in germinated barnyard millet [12]. The optimal soaking time, germination temperature, and germination time were 11.78 h, 33 °C, and 36.48 h, respectively, and the corresponding total phenolic content (TPC) and total flavonoid content (TFC) were 54.99 mg GAE/g and 42.56 mg RUE/g, respectively. Sharma et al. optimized the germination conditions for kodo millet by central composite rotatable design, and the TPC and TFC of the germinated kodo millet flour were 83.01 mg GAE/100 g and 87.53 mg RUE/g, respectively, under optimal conditions [10]. On the other hand, Paucar-Menacho et al. found that the total phenolic content (TPC) of quinoa significantly increased by 80%, when optimally sprouted at 20 °C for 42 h [13]. When contrasted with the control (without NaCl solution treatment), Lim et al. discovered that the amount of phenolic compounds and carotenoids in buckwheat sprouts increased to different extents when treated with varying concentrations of NaCl solution (10, 50, 100, and 200 mM) [14]. Brown rice treated with 50–200 mM CaCl_2_ solution during germination showed a significantly higher accumulation of bioactive compounds in a concentration-dependent manner, such as polyphenols, flavonoids, and GABA [15]. According to Nguyen et al., a 50 mM sucrose solution increased the concentration of total flavonoids, anthocyanins, and total phenolics in sprouted colored brown rice [16].

Our previous research showed that germination could significantly increase the TPC and GABA content of foxtail millet [17]. However, the optimal germination conditions for the simultaneous enrichment of GABA and polyphenols have not been reported. Therefore, in order to effectively increase phenolic and GABA contents and improve the health benefits of foxtail millets, germination conditions and culture solutions for foxtail millet sprouts were optimized by screening single factors and applying response surface methodology (RSM). In addition, the individual phenolic compounds were also investigated, which should provide theoretical basis to improve bioactive components in foxtail millet sprouts and to increase added value of foxtail millets.

## 2. Materials and Methods

### 2.1. Chemicals and Reagents

Methanol (MeOH), sodium carbonate (Na_2_CO_3_), hexane, ethyl acetate, sodium nitrite (NaNO_2_), sodium hydroxide (NaOH), aluminum chloride (AlCl_3_), iron (III) chloride (FeCl_3_), sucrose, calcium chloride (CaCl_2_), and sodium chloride (NaCl) were all acquired from Tianjin Dean Chemical Reagent Co., Ltd. (Tianjin, China). Folin–Ciocalteu reagent was purchased from Sigma-Aldrich (St. Louis, MO, USA). HPLC and mass-spectrometry-grade methanol and formic acid were purchased from Thermo Fisher Scientific Reagent Co., Ltd. (Waltham, MA, USA). The standards, γ-aminobutyric acid (GABA), protocatechuic aldehyde, *p*-coumaric acid, *p*-hydroxybenzoic acid, *tran*-ferulic acid, *p*-hydroxybenzaldehyde, protocatechuic aldehyde, and protocatechuic acid, were purchased from Shanghai Yuanye Biotechnology Co., Ltd. (Shanghai, China). Except for the specially marked chemical reagent grade, other reagents were all of analytical grade.

### 2.2. Material and Sprouting Process

Foxtail millet samples were harvested in September 2022 from farmland in Yichuan County, Luoyang City, and provided by Jinsu Agricultural Technology Co., Ltd., Luoyang City, Henan Province, China.

Foxtail millet was immersed in pure water for a period of time after being screened for any impurities. Briefly, the seeds were equally distributed in Petri dishes lined with three layers of filter paper after the excess water was drained and germinated in a constant-temperature incubator for a period of time. Daily, the germinated samples were collected, freeze-dried, and subsequently pulverized to pass through a 40-mesh sieve and stored at −20 °C before being analyzed.

### 2.3. Extraction and Measurement of Polyphenols

The free and bound phenolic compounds were obtained from foxtail millet sprouts using the extraction method previously reported [18]. Briefly, 1 g of the germinated foxtail millet sample was defatted with N-hexane (1:10 *w/v*) for 15 min, and then the defatted sample was extracted twice with 80% methanol for 1 h each time. The supernatant was combined as the free phenolic extract. The residue after the extraction of free phenolic was dried and then digested with 2 mol/L NaOH solution for 2 h under dark. The hydrolysate was adjusted to pH 1.5–2.0 with 6 mol/L HCl. After centrifugation, the hydrolysate was extracted with ethyl acetate three times to obtain the bound phenolic extract. The free and bound phenolic extracts were evaporated by a rotary evaporator at 40 °C (RE52CS, Shanghai Yarong Biochemical Instrument Factory, Shanghai, China) and then redissolved in 50% methanol to obtain the test solution. The germinated foxtail millet’s TPC was measured using a 96-well microplate colorimetric method [19]. The free and bound TPC were expressed as milligrams of ferulic acid equivalents per 100 g (mg FAE/100 g DW) using a ferulic acid standard curve. The TPC optimized by single-factor analysis and response surface optimization was the sum of free and bound phenolic fractions.

UPLC-MS/MS was employed to identify and quantify individual phenolic compounds. The chromatography was conducted using an Accucore C_18_ column (100 mm × 3 mm, Thermo Fisher Scientific, Waltham, MA, USA), with a temperature of 30 °C and an injection volume of 10 µL. Our previously reported method [20] was followed for the gradient elution procedure and mass spectrometer parameters. The external standards method was employed to quantify the content of each phenolic compound in milligrams per kilogram (mg/kg).

### 2.4. Extraction and GABA Measurement

MeOH solution (80%) was employed to extract GABA, as previously mentioned in the extraction of free phenolics [21]. The same method as the free phenolic extraction procedure described above was followed. The column and UPLC-MS equipment were the same as those used for phenolic analysis, with the column temperature maintained at 30 °C and an injection volume of 10 µL. The mobile phase was A: water (0.1% formic acid) and B: acetonitrile, operating at a flow rate of 0.2 L/min. The elution protocol consisted as follows: 0–3 min, 95–5% B; 3–4 min, 10–90% B; 4–6 min, 10–90% B; and 6–8 min, 95–5% B. In positive ion (ESI ^+^) mode, multiple reaction monitoring (MRM) was employed, using a capillary voltage of 0.35 KV, a cone hole voltage of 30 V, an ion source temperature of 150 °C, and a dissolvent gas temperature of 500 °C. The flow rates of the cone hole gas (He) and the dissolvent gas (N_2_) were set at 20 L/h and 800 L/h, respectively. The quantitative ion pairs were established at 104.0/87.0 (*m/z*). The GABA concentration in sprouted foxtail millet samples was quantified using an external standard.

### 2.5. Experimental Design for Optimizing Germination Conditions

#### 2.5.1. Soaking Conditions

Soaking time and temperature were examined independently. The selected foxtail millet grains were soaked in pure water at temperatures ranging from 15 to 25 °C for a period from 2 to12 h, maintaining 90% humidity, and germinated at 25 °C for 3 days (d). TPC and GABA content of the sprouted foxtail millet samples were then determined.

#### 2.5.2. Germination Conditions

Germination time and temperature were examined independently. Foxtail millet samples were soaked in pure water at 30 °C for 4 h, pure water was applied to maintain 90% humidity, and germination was carried out at temperatures ranging from 15 to 35 °C for 1–5 d, respectively. The foxtail millet sprouts were collected to determine TPC and GABA content.

#### 2.5.3. Culture Media Solutions

After being soaked in pure water at 30 °C for 4 h, different culture solutions—NaCl concentration: 0 mM, 20 mM, 40 mM, 60 mM, 80 mM, and 100 mM; CaCl_2_ concentration: 0 mM, 2 mM, 4 mM, 6 mM, and 8 mM; sucrose concentration: 0 g/L, 1 g/L, 5 g/L, 9 g/L, 13 g/L, and 17 g/L; 5 g/L sucrose + CaCl_2_: 0 mM CaCl_2_, 2 mM CaCl_2_, 4 mM CaCl_2_, 6 mM CaCl_2_, and 8 mM CaCl_2_, and 6 mM CaCl_2_+ sucrose: 1 g/L sucrose, 5 g/L sucrose, 9 g/L sucrose, 13 g/L sucrose, and 17 g/L sucrose—were used to maintain 90% humidity, independently. Following, TPC and GABA content of foxtail millet sprouts were measured.

#### 2.5.4. Response Surface Methodology (RSM)

The RSM experiment was developed using a 3-factor, 3-level Box–Behnken approach to investigate the effects of these independent factors on the TPC and GABA levels, as shown in Table 1. The three independent variables were soaking time (X_1_), soaking temperature (X_2_), and sucrose concentration (X_3_). In addition, the TPC and GABA levels in the extracts from foxtail millet sprouts were considered as response variables.

Design Expert 8.0.6 software (11.0.1.0 64-bit, State-East Corporation, Los Angeles, CA, USA) was employed to calculate regression equations. The second-order polynomial model was used to derive regression coefficients from the experimental findings (Equation (1)). The F-value, *p*-value, R^2^ value (coefficient of determination), R^2^_adj_ value (adjusted coefficient of determination), and R^2^pred value (predicted coefficient of determination) of the constructed models were assessed using the reflecting parameters [22]. The optimal input variable values were determined using a desirability function approach to optimize the yields of TPC and GABA. Subsequently, the theoretical findings were validated by performing triplicate extractions under optimal conditions, with a comparison of the experimental results to the projected TPC and GABA content.
(1)$$\gamma =\beta_{0}+\sum_{j=1}^{k}\beta_{j}x_{j}+\sum_{j=1}^{k}\beta_{jj}x_{j}^{2}+\sum_{i}\sum_{<j=2}^{k}\beta_{jj}x_{j}x_{j}+e_{i}$$

### 2.6. Statistical Analysis

Results are presented as mean ± standard deviation, and all experiments were carried out in triplicates. The results data were subjected to an analysis of variance (ANOVA) in IBM SPSS Statistics (v.25.0, IBM Corp., Armonk, NY, USA). In the ANOVA model, all data were consistent with homogeneity of variance and normality. Subsequently, Duncan’s multiple range test and *t*-tests to ascertain statistically significant differences were employed. The confidence interval of each treatment group was set to 95%, and statistical significance was defined as *p* < 0.05.

## 3. Results and Discussion

### 3.1. Effects of Soaking Conditions on GABA and TPC of Foxtail Millet Sprouts

#### 3.1.1. Soaking Time

The effect of soaking time on the GABA content of the germinated foxtail millets is shown in Figure 1A. It can be seen the GABA in foxtail millets showed an overall trend of first increasing and then decreasing with the prolongation of soaking time, reaching the maximum value of 159.90 mg/kg at soaking for 4 h. This result is similar to the trend of GABA in germinating adzuki beans [23]. This may be because the dry matter in the endosperm at the early stage of soaking is converted into soluble matter through actions of enzymes, supplying nutrients needed for endosperm respiration and germination and laying the foundation for production of GABA, so that the GABA content in the grains rises with relatively shorter soaking time. However, when the soaking time is prolonged, the cell structure is damaged, a significant amount of water-soluble substances will leach out, and the amount of substrate for generating GABA is reduced, so that GABA in the germinated grain seeds decreases [24].

The effect of soaking time on the TPC of foxtail millets is shown in Figure 1B. The TPC of foxtail millet sprouts shared a similar trend of first increasing and then decreasing with GABA, and it reached the maximum value of 660.71 mg FAE/100 g DW after being soaked for 4 h. This performance is similar to the result of Li et al., who found that the polyphenol content of mung beans increased firstly and then decreased with the extension of soaking time [25]. The trends of free phenolic content and bound phenolic content shared a similar trend to the TPC sum. The polyphenol synthase in foxtail millet started to be activated at the beginning of soaking, and soaking process also softened structure of seeds, allowing the release of phenolics bound to the cell wall through ester and ether bonds, which led to the increase in polyphenol content at the beginning of soaking. However, as the soaking time increased, a portion of free phenolics in the grains were leaching out, which caused a decrease in the total phenolic content with the prolonged soaking period [5].

#### 3.1.2. Soaking Temperature

The effects of soaking temperature on the GABA content and TPC of foxtail millet sprouts are shown in Figure 1A,B. The GABA content and TPC of foxtail millet sprouts increased significantly (*p* < 0.05) at first and then decreased with the increase of the soaking temperature. The trends of free phenolic content and bound phenolic content were similar to that of the TPC sum. Both GABA content and TPC sum reached maximum values of 275.17 mg/kg and 758.54 mg FAE/100 g DW, respectively, when soaking at 30 °C, which were 93.69% and 17.32% higher than those values of soaking at 15 °C. However, significant decreases in phenolic and GABA content were observed when the immersion temperature exceeded 30 °C, which may have been because the relatively high soaking temperature altered the rate of uptake of the grains as well as the activity of enzymes, such as glutamic acid decarboxylase (GAD) and phenylalanine deaminase (PAL) associated with the synthesis of GABA and phenolic compounds. When the soaking temperature increased, the water absorption rate of foxtail millet seed was accelerated, and large molecules within the seed were decomposed into small molecules under the action of hydrolases, which provided sufficient material basis for the production of GABA and polyphenols, which were accumulated under the actions of PAL and GAD [26,27]. As the soaking temperature rose, the water absorption rate of grain seeds grew faster, and the cell structure may have been damaged when the seeds were saturated with water, leading to the leaching of glutamic acid, phenylalanine, and other water-soluble substances [28]. Similar results were reported in studies on germination of barley and wheat by Singkhornart and Ryu [29].

### 3.2. Effect of Germination Conditions on GABA and TPC of Foxtail Millet Sprouts

#### 3.2.1. Germination Time

The effects of germination time on the GABA and TPC of foxtail millets are presented in Figure 2A,B. Within the selected range of germination time, GABA in foxtail millet sprouts increased with germination time and displayed the maximum value of 756.86 mg/kg after germination for 5 d. The TPC presented similar trend with GABA content, with the highest at 4 d and 5 d without significant difference. A similar performance was also observed in the germination of wheat [7], barley, and brown rice [30]. Our previous research also revealed this trend in the germination of foxtail millets [17]. This was because GAD and PAL were activated and remained active during germination under the action of synthetic enzymes, causing GABA and polyphenols to be accumulated in the germination process as the time was prolonged [31].

#### 3.2.2. Germination Temperature

The effects of germination temperature on the GABA and TPC of foxtail millets are shown in Figure 2A,B. The GABA level and TPC of foxtail millet sprouts increased with increasing temperature in the selected experimental temperature range under the conditions of soaking at 30 °C for 4 h, reaching 677.56 mg/kg and 919.07 mg FAE/100 g DW, respectively, at a germination temperature of 35 °C. This performance was attributed to the increase in activities of GAD and PAL with the increasing germination temperature, which provided the basis for the accumulations of GABA and phenolics. In addition, the increase in germination temperature within the appropriate range affected the germination cycle of foxtail millet grain, with significant differences in the accumulation of GABA and phenolics as germination cycle changed [10].

### 3.3. Effect of Culture Media on GABA and TPC of Foxtail Millet Sprouts

#### 3.3.1. Effects of NaCl Concentration on GABA and TPC

The effects of different concentrations of NaCl solution on the GABA and TPC of foxtail millet sprouts are shown in Figure 3A,B. With the increase in NaCl concentration, both the GABA level and TPC showed firstly increasing and then decreasing trends. The GABA content reached the maximum of 156.47 mg/kg at 20 mM NaCl concentration, which was 4.03% higher than the control (foxtail millet germinated with pure water). The changes in TPC with the NaCl solution treatment were similar to those of GABA, reaching 832.98 mg FAE/100 g DW in the foxtail millet sprouts at a 20 mM NaCl solution, which was 14.09% higher than the control.

The relatively low-concentration NaCl solution significantly promoted the accumulation of GABA and phenolics, which might have been because the low-NaCl solution increased GAD activity, which led to a higher accumulation of GABA. Moreover, GABA is a signaling molecule that has been extensively studied in plant stress signaling. GABA should increase the activities of PAL and cinnamic acid 4-hydroxylase (C4H), which affect metabolism and synthesis of phenolics in plants [32]. Similar performances were exhibited in germination of buckwheat [33], wheat [34], and oilseed rape seeds, where moderate NaCl stress effectively enriched polyphenols and GABA in these grain sprouts.

#### 3.3.2. Effects of CaCl_2_ Concentration on GABA and TPC

The effects of CaCl_2_ solution on the GABA and TPC of foxtail millet sprouts are shown in Figure 3A,B. CaCl_2_ significantly influenced the GABA and TPC of the sprouted millets, and the GABA content and TPC tended to increase and then decrease in the selected range of CaCl_2_ concentration from 0 to 8 mM. The highest levels of phenolics and GABA were reached at 6 mM CaCl_2_ concentration, with 906.89 mg FAE/100 g DW and 183.78 mg/kg, respectively, which were increased by 24.21% and 20.81%, respectively, comparing to the control.

It is well known that GABA formation is mainly attributed to the decarboxylation of L-glutamate in grains, caused by GAD being activated during germination process. The exogenous application of CaCl_2_ activates GAD, which results in the accumulation of GABA [35]. In return, GABA should increase the activity of PAL, and the addition of exogenous Ca^2+^ during seed germination has a significant effect on activity of PAL, which is crucially involved in synthesis of phenolics in grains [36]. CaCl_2_ treatment was found to increase polyphenol content in mung bean sprouts [25]. Additionally, under abiotic stress conditions, CaCl_2_ is essential for mitigating plant damage and initiating cellular repair by controlling various cellular processes, such as lipid peroxidation and antioxidant enzyme activities [37].

#### 3.3.3. Effects of Sucrose Solution Concentration on GABA and TPC

The different concentrations of sucrose solution on the GABA in foxtail millets is shown in Figure 3A. The GABA content of millet sprouts reached 192.71 mg/kg at the sucrose concentration of 5 g/L, which was 26.67% higher than the control. This may have been due to the increased activity of GAD when the sucrose concentration was relatively low, leading to the accumulation of GABA. With the increase of sucrose concentration, the cellular osmotic pressure should be altered, along with the disruption of cellular structures, and high sucrose concentrations also subject foxtail millet grains to stress effects, which lead to the inhibition of GABA synthesis. This result is similar to the findings of Jeong et al. on buckwheat germinated with sucrose solutions [38], and Nguyen et al. [16], who found that polyphenols and flavonoids in brown rice sprouts could be significantly increased by suitable sucrose concertation treatment.

The different concentrations of sucrose solution on the TPC of foxtail millet sprouts is shown in Figure 3B. The TPC of foxtail millet sprouts reached 922.59 mg FAE/100 g DW at the sucrose concentration of 5 g/L, which was 26.36% higher than the control. This may be have been due to the increase in phenolic derivatives resulting from the build-up of sucrose in the foxtail millet germination process through the activation of sugar metabolism [39]. Research has shown that exogenous sucrose may be responsible for the accumulation of antioxidant compounds, such as ascorbic acid and polyphenols, by upregulating the expression of relevant genes [40].

#### 3.3.4. Effects of Combination of Sucrose and CaCl_2_ on GABA and TPC

The effects of the combinations of 5 g/L sucrose and different concentrations of CaCl_2_ solution on GABA and TPC of foxtail millet sprouts are shown in Figure 3A,B. At constant sucrose concentration, the GABA in foxtail millet sprouts showed an increasing and then decreasing trend with the increase of the CaCl_2_ concentration. The maximum GABA level of 185.98 mg/kg was achieved under the combination culture of 5 g/L sucrose and 6 mM CaCl_2_, which was not significantly different from that under 5 g/L sucrose or 6 mM CaCl_2_ alone for germination. The trend of TPC was similar to that of GABA, and no significant difference was found between TPC under the combination culture of 5 g/L sucrose and 6 mM CaCl_2_ and germination solely with 5 g/L sucrose solution.

The effects of 6mM CaCl_2_ in combination with different concentrations of sucrose solution on GABA and TPC of foxtail millet sprouts are shown in Figure 3A,B. The GABA in foxtail millet sprouts showed an overall trend of increasing and then decreasing with increasing sucrose concentration at a constant CaCl_2_ concentration of 6 mM. When the combined solution was composed of 6 mM CaCl_2_ and 5 g/L sucrose, the GABA content of foxtail millet sprouts reached 172.17 mg/kg, which was significantly lower than the value of 192.71 mg/kg when 5 g/L sucrose solution alone was used as culture medium for germination. For the combined solution of culture media, the increase of sucrose concentration surpassing 5 g/L concertation resulted in a decrease in GABA in the foxtail millet sprouts. The performance for TPC was similar to that for GABA, with the maximum TPC in foxtail millet sprouts when the combination of 6 mM CaCl_2_ and 5 g/L sucrose was used as the culture medium, but no significant difference was achieved comparing with 5 g/L sucrose or 6 mM CaCl_2_ alone as the culture medium for germination. Therefore, the 5 g/L sucrose solution may be the better culture medium for the production of enriched phenolics and GABA foxtail millet sprouts.

### 3.4. Model Fitting

The results of 17 improvements of phenolics and GABA based on various germination conditions are presented in Table 2. The regression equations and quadratic polynomial model of yield were tested for fitting using *t*-tests and ANOVA, respectively [22]. The *p*-value indicated the relevance of the variables; a smaller *p*-value denoted a more substantial influence of the variable reflected on the outcomes [41]. The F-values were employed to evaluate the relative contributions of each component to the TPC and GABA yields, and the findings are shown in Table 3. Additionally, Figure 4 illustrates the 3D plots and associated contour plots generated from the model, effectively demonstrating the impact of substantial interaction factors on the responses of TPC and GABA levels. The response value was influenced by the interactive factors, as indicated by the inclination of the surface in the 3D diagram. The interaction between these two factors was more significant as the inclination increased. However, the values on each curve in the contour plots of the response surface were identical. The graph’s color transitioned from blue to red, denoting increased values.

#### 3.4.1. Total Polyphenol Content

As shown in Table 3, the *p*-value of the regression model for the TPC response values was less than 0.001, indicating that the regression model was statistically significant [42]. The F-value of the lack of fit was 0.43, and the *p*-value was greater than 0.05, indicating that the lack of fit was not significant, and the regression equation fitted well. The coefficient of determination (R^2^) of the regression equation was 0.9866, with a correction coefficient (R^2^_adj_) of 0.9693. The closer the R^2^ value is to 1 and the closer the R^2^_adj_ value is to R^2^ values are indicators of the goodness of fit of the model, and the difference between the two was less than 0.2, which indicated that the model was good fit and could be used to predict the relationship between TPC and soaking time, soaking temperature, and sucrose concentration, as well as to optimize the germination conditions for the enrichment of phenolics in foxtail millet sprouts. The coefficient of variation (CV) of the model was 0.98%, which also indicates that the model had a small degree of variation and a high confidence level.

It could be seen that the *p*-value of less than 0.05 for soaking time (X_1_) was significant. Soaking temperature (X_2_), and sucrose concentration (X_3_) were highly remarkable, contributing to the model, and displayed marked effect on TPC. The *p*-values of the quadratic terms X_1_^2^, X_2_^2^, and X_3_^2^ were less than 0.001, which were highly significant, contributed to the model, and presented significant effects on TPC. The *p*-value of the interaction term was less than 0.05 only for X_2_ × X_3_, which was shown to be significant by the *p*-value test, indicating that only X_2_ × X_3_ in the interaction term had a significant effect on TPC. Whereas the other interactions did not have significant effects on enrichment of phenolics in foxtail millet sprouts (*p* > 0.05). The data were analyzed using multiple regression by removing the non-significant factors to obtain the final equation for TPC of the foxtail millet sprouts, as shown in Equation (2).
TPC = 927.24 + 9.89 × X_1_ + 12.83 × X_2_ − 23.72 × X_3_ + 10.50 × X_2_ × X_3_ − 42.86 × X_1_^2^ − 46.49 × X_2_^2^ − 47.36 × X_3_^2^(2)

The interaction between soaking time and soaking temperature is shown in Figure 4a, and interaction between soaking time and sucrose concentration is displayed in Figure 4b; both of them had no significant effect (*p* > 0.05) on the TPC of foxtail millet sprouts. As shown in Figure 4c,d, a highly significant interaction (*p* < 0.05) between soaking temperature and sucrose concentration on TPC was achieved, showing a tendency of first increasing and then decreasing when the soaking temperature was varied at the constant sucrose concentration. On the other side, the TPC of foxtail millet sprouts showed an initial increase and then decreasing trend with increasing sucrose concentration at the constant soaking temperature. When the soaking time was constant, the soaking temperature was 30 °C, and the sucrose concentration was 5 g/L as the highest point of the response surface; the phenolic enrichment of foxtail millet sprouts reached the maximum of 935.93 mg FAE/100 g DW, and the further increase of these two factors would negatively affect the TPC [16].

#### 3.4.2. GABA Content

As can be seen in Table 3, the *p*-value of the regression model for the GABA response value was less than 0.001, which showed high significance. The F-value of the lack of fit was 1.01 and the *p*-value was greater than 0.05, indicating that the lack of fit was not significant, suggesting that the regression model was statistically significant and could be used to predict GABA content of foxtail millet sprouts. The coefficient of determination (R^2^) of the regression equation was 0.9876, while the corrected coefficient (R^2^_adj_) was 0.9716, and the difference was less than 0.2, indicating that the model fitted well and could be used for predicting the relationship between GABA content and soaking time, soaking temperature, and sucrose concentration, as well as for optimizing the germination conditions for the enrichment of GABA in foxtail millet sprouts. The coefficient of variation (CV) of the model was 1.41%, which also indicates that the model had a small degree of variation and a high confidence level. It could be seen from Table 3 that the *p*-value of one item, soaking time (X_1_), and soaking temperature (X_2_), which was less than 0.001, were highly significant, contributing to the model and presented a significant effect on the GABA. The *p*-value of sucrose concentration (X_3_) was greater than 0.05 and did not have significant effect on GABA. The *p*-values of the secondary terms X_1_^2^, X_2_^2^, and X_3_^2^ were less than 0.001, which showed highly significant and contributed significantly to the model, with a significant effect on GABA. Only X_1_ × X_3_ of the interaction term had *p*-value less than 0.05, which was shown to be significant, indicating that only X_1_ × X_3_ of the interaction term had a significant effect on GABA, and the other interactions did not show significant effect on GABA enrichment of foxtail millet sprouts (*p* > 0.05). The data were analyzed using multiple regression by removing the non-significant factors to obtain the final equation for the enrichment of GABA in foxtail millet sprouts, as shown in Equation (3).
GABA = 261.01 + 9.59 × X_1_ + 9.78 × X_2_ − 4.33 × X_1_ × X_3_ − 10.86 × X_1_^2^ − 23.80 × X_2_^2^ − 17.30 × X_3_^2^(3)

The interaction between soaking time and soaking temperature is shown in Figure 5a, and interaction between soaking temperature and sucrose concentration is displayed in Figure 5c; both of them had no significant effect (*p* > 0.05) on the GABA enrichment of foxtail millet sprouts. A significant (*p* < 0.05) interaction between soaking time and sucrose concentration was achieved and is exhibited in Figure 5b,d. The GABA content of foxtail millet sprouts increased with the increasing soaking time and sucrose concentration, reaching the maximum value and then starting to decrease, suggesting that too long of a soaking time and excessive sucrose concentration could negatively affect the enrichment of GABA in foxtail millet sprouts [5,40].

#### 3.4.3. Experimental Validation of the RSM Mode

Based on the results of the response surface test and calculated by the regression models, the maximum predicted values of TPC and GABA content of foxtail millet sprouts were 929.69 mg FAE/100 g DW and 263.54 mg/kg, respectively, which corresponded to the optimal germination conditions: soaking time of 4.5 h, soaking temperature of 30.83 °C, and sucrose concentration of 4.49 g/L. Considering the workability of the experiment design, the conditions for germination of foxtail millet were modified to 4.5 h of soaking time, 31 °C of soaking temperature, and 4.5 g/L of sucrose concentration. The maximum predicted values of the TPC and GABA content of foxtail millet sprouts were 929.44 mg FAE/100 g DW and 263.60 mg/kg, respectively, which corresponded to the modified germination conditions: soaking time of 4.5 h, soaking temperature of 31 °C, and sucrose concentration of 4.5 g/L. Three replicates of the modified germination conditions yielded the foxtail millet sprouts with TPC of 926.53 ± 1.23 mg FAE/100 g DW and GABA content of 259.13 ± 1.83 mg/kg, which were small relative errors from the pre-modification results, indicating this model was reliable. The fitting degree of TPC and GABA with the model reached 99.69% and 98.30%, respectively, and the fitting effect was extremely good. The free TPC, bound TPC, TPC sum, and GABA level increased by 156.01, 52.23, 84.81, and 6%, respectively, compared to the pre-optimized germination conditions. The optimized free TPC, bound TPC, TPC sum, and GABA levels were increased by 3.57-, 1.49-, 1.99-, and 9.77-fold, respectively, compared to the ungerminated samples. This may have been due to the further increase of phenylalanine deaminase activity and enhanced metabolism of phenylpropane metabolic pathway in foxtail millet sprouts under optimal conditions, which firstly generated trans-cinnamic acid under the action of phenylalanine ammonia-lyase (PAL), and then biosynthesized flavonoids, phenolic acids, and other phenolics under the action of cinnamic acid 4-hydroxylase (C4H) and 4-hydroxylase [43]. The activity of glutamic acid decarboxylase, a key metabolizing enzyme of GABA, could be increased, resulting in a rapid accumulation of GABA, while the activity of GABA transaminase (GABA-T) may be decreased and the rate of GABA degradation reduced under optimal germination conditions [44]. The present study found that GABA and most of phenolic compounds in foxtail millet were greatly accumulated under the optimized germination conditions. However, the reasons for the enrichments of these bioactive compounds are undiscovered. Future studies should be carried out to explore the changes in related enzyme activities during millet germination in order to reveal the underlying mechanisms of enrichments in GABA and phenolic compounds.

### 3.5. Comparison on the Individual Phenolic Levels Before and After Optimization of Germination Conditions

Phenolics during the germination of foxtail millet had been characterized in our previous study [17]. We carried out a comparative analysis of the main individual phenolics of foxtail millet sprouts before and after the optimization of germination conditions, and the results are shown in Table 4. Both free and bound individual phenolic compounds showed significant differences between before and after optimization of the germination conditions. Among the 11 free phenolics, feruloyl quinic acid, *trans*-ferulic acid, and 3,7-dimethyl quercetin were most abundant and increased by 2.11, 1.84, and 0.19 times, respectively, by the optimized germination conditions. The two rare 5-hydroxytryptamine phenolic derivatives in cereal grains, namely N-(*p*-coumaroyl) serotonin, N-feruloylserotonin, increased by 4.13-fold and 4.11-fold, respectively. And 1-*O*-*p*-coumaroyl-3-*O*-ferulic acid glycerol increased from 4.47 mg/kg DW to 38.31 mg/kg DW.

Trans-*p*-coumaric acid and trans-ferulic acid, which were the most abundant polyphenols in foxtail millet sprouts predominant in bound, increased from 1167.33 mg/kg DW to 1386.38 mg/kg DW and from 2325.40 mg/kg DW to 4032.13 mg/kg DW, respectively, after optimization of the germination conditions. However, their corresponding *cis* isomers with comparably lower levels in foxtail millet sprouts, namely *cis*-*p*-coumaric acid and *cis*-ferulic acid, showed significant decreases of 19.41% and 34.45%, respectively. It may be attributed to the fact that the synthesis of *cis*-*p*-coumaric acid and *cis*-ferulic acid were inhibited during the optimized germination process, or a proportion of *cis* isomers were transformed to their corresponding *trans* isomers. Except for these two cis hydroxycinnamic acids, all the remaining bound phenolic derivatives showed significant increase by the optimization of germination conditions. The interaction of soaking temperature, soaking time, and sucrose concentration may increase the activity of phenylalanine ammonia lyase, which is related to polyphenol synthesis, resulting in an increase in the content of phenolic derivatives.

Our optimized germination conditions possessed a shorter soaking time, slightly higher soaking temperature, and addition of sucrose as culture phytonutrient, which may have reduced the leaching of water-soluble phenolics and further increased the activity of phenolic synthases [16], which provided the basis for the enrichment of phenolic compounds, leading to significant increases in most of phenolics in foxtail millet sprouts [10,45].

## 4. Conclusions

The germination conditions for the enrichment of phenolics and GABA in foxtail millet were screened, and the three significant factors were identified as soaking time, soaking temperature, and sucrose concertation of culture solution. The response surface method was applied to optimize the germination conditions for foxtail millet sprouts as follows, soaking for 4.5 h at 31 °C and germinating with 4.5 g/L sucrose solution for 5 days at 35 °C. Under the optimal germination conditions, TPC and GABA content reached 926.53 mg FAE/100 g DW and 259.13 mg/kg, respectively. Under the same conditions, the predicted values of TPC and GABA were 929.44 mg FAE/100 g DW and 263.60 mg/kg, respectively, which were less different from the actually measured values. Moreover, except for bound *cis*-*p*-coumaric acid and *cis*-ferulic acid, all the free and bound individual phenolic compounds were significantly increased under the optimal germination process, especially for the predominate *trans*-*p*-coumaric acid and *trans*-ferulic acid in bound. The optimized germination conditions could be carried out to produce polyphenols and GABA simultaneously enriched foxtail millet sprouts, which could be added as a functional ingredient in food products such as dough, bread, and pasta and also could be consumed as healthy microgreens.

## Figures and Tables

**Figure 1 foods-13-03340-f001:**
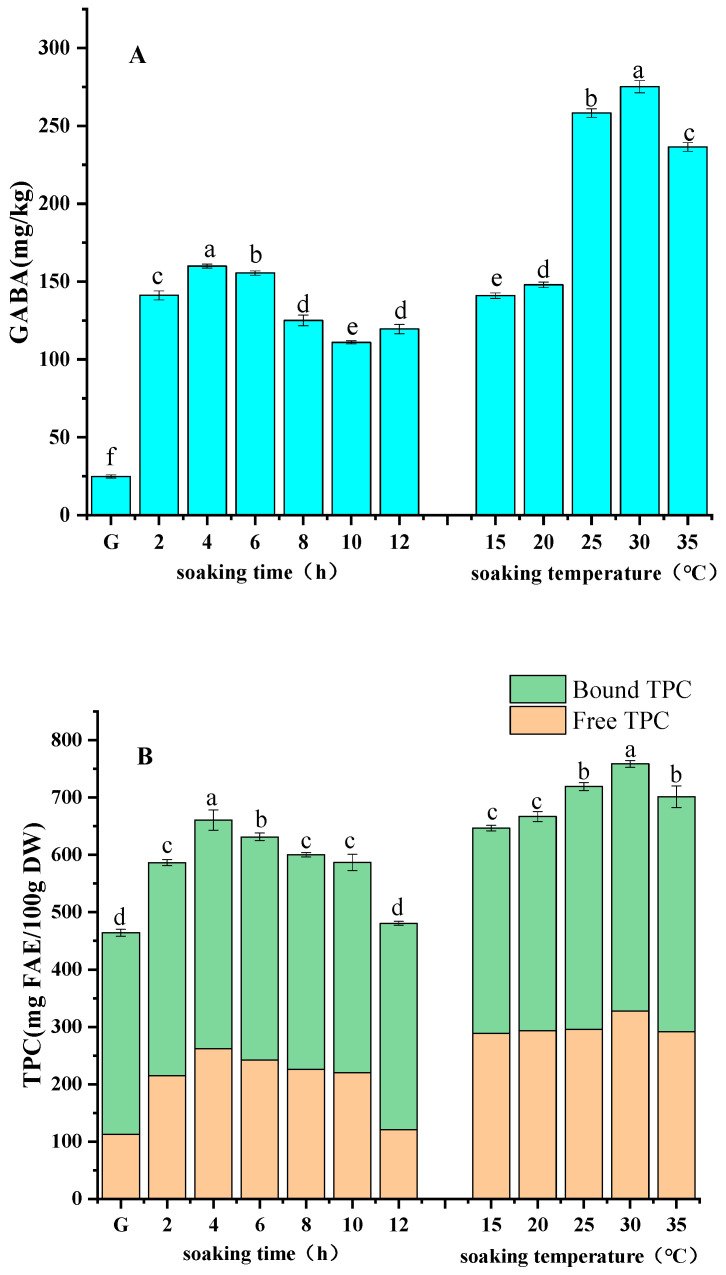
Effects of soaking conditions (soaking time and temperature) on the GABA content (**A**) and TPC (**B**) of germinated foxtail millets. G: ungerminated foxtail millet seeds. Different lowercases in the same histogram indicate statistically significant differences (*p* < 0.05).

**Figure 2 foods-13-03340-f002:**
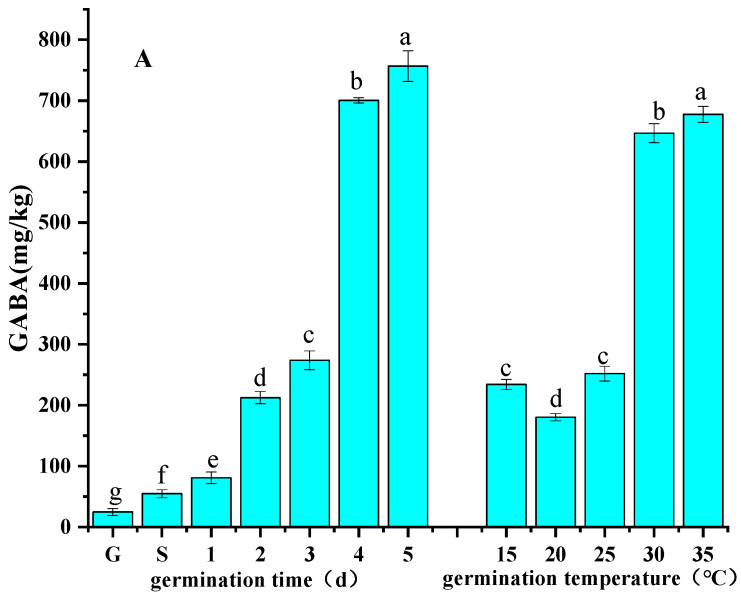
Effects of germination conditions (germination time and temperature) on the GABA content (**A**) and TPC (**B**) of germinated foxtail millets. G: ungerminated foxtail millet seeds, S: presoaked foxtail millet seeds. Different lowercases in the same histogram indicate statistically significant differences (*p* < 0.05).

**Figure 3 foods-13-03340-f003:**
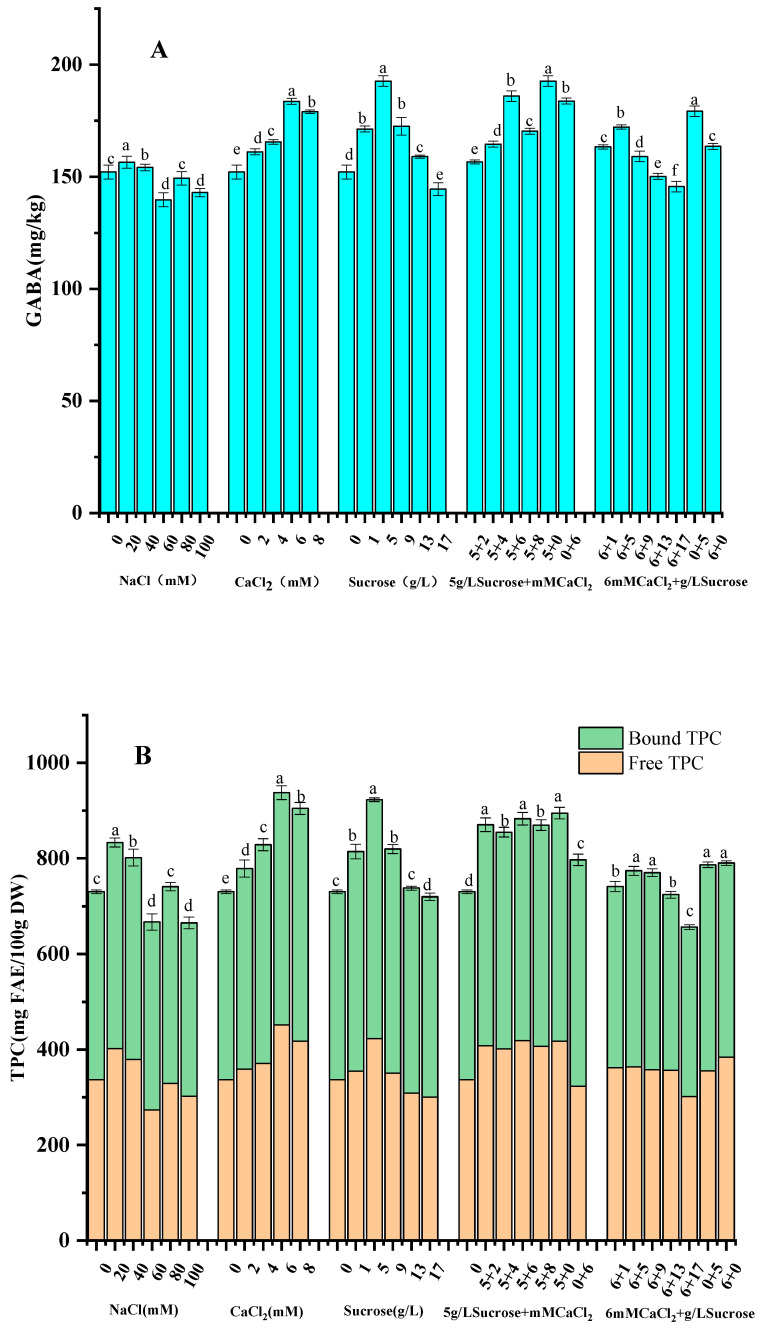
Effects of culture solutions on GABA content (**A**) and TPC (**B**) of germinated foxtail millets. Different lowercases in the same histogram indicate statistically significant differences (*p* < 0.05).

**Figure 4 foods-13-03340-f004:**
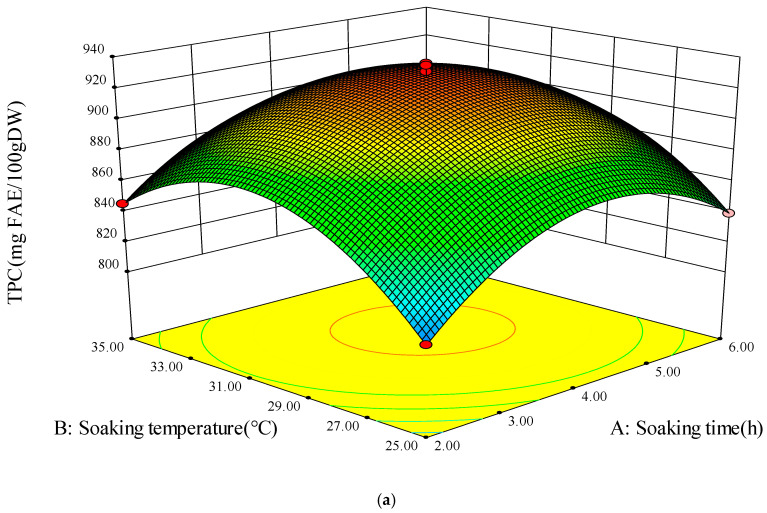
Three-dimensional response surface plots and corresponding contour plots. Interactions of soaking time and soaking temperature (**a**), soaking time and sucrose concentration (**b**), and soaking temperature and sucrose concentration (**c**,**d**) on TPC.

**Figure 5 foods-13-03340-f005:**
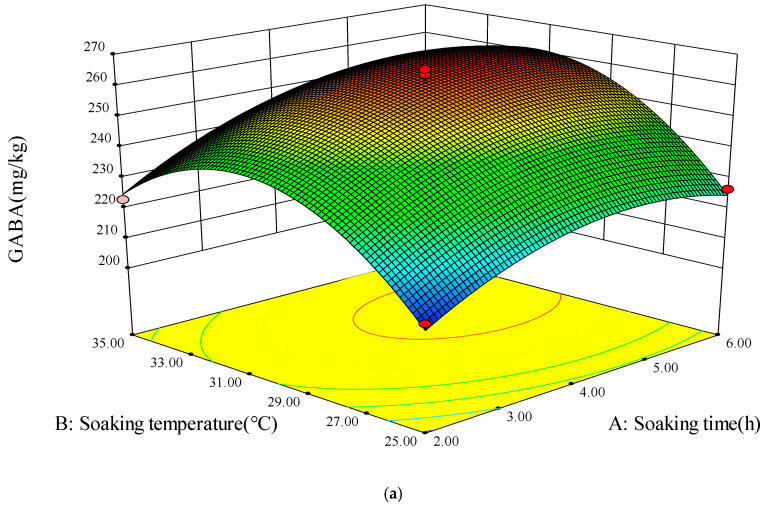
Three-dimensional response surface plots and corresponding contour plots. Interactions of soaking time and soaking temperature (**a**), soaking time and sucrose concentration (**b**,**d**), and soaking temperature and sucrose concentration (**c**) on GABA.

**Table 1 foods-13-03340-t001:** Experimental design independent variables and their levels.

Levels	Independent Variables		
	Soaking Time (h)	Soaking Temperature (°C)	Sucrose Concentration (g/L)
−1	2	25	1
0	4	30	5
1	6	35	9

**Table 2 foods-13-03340-t002:** Experimental design and results of the response surface methodology.

	Variables				
Run	X_1_ (h)	X_2_ (°C)	X_3_ (g/L)	TPC (mg FAE/100 gDW)	GABA (mg/kg)
1	2	25	5	810.85 ± 1.54	210.52 ± 0.12
2	6	25	5	839.28 ± 2.18	226.45 ± 0.25
3	2	35	5	845.63 ± 0.89	223.00 ± 0.28
4	6	35	5	855.80 ± 1.23	245.42 ± 0.56
5	2	30	1	855.63 ± 0.55	217.75 ± 1.01
6	6	30	1	873.63 ± 0.83	245.60 ± 0.26
7	2	30	9	798.15 ± 0.78	228.76 ± 0.31
8	6	30	9	820.67 ± 0.05	239.27 ± 0.45
9	4	25	1	850.89 ± 0.23	209.43 ± 0.55
10	4	35	1	855.54 ± 1.67	230.90 ± 0.24
11	4	25	9	790.24 ± 1.34	207.01 ± 0.10
12	4	35	9	836.89 ± 0.67	232.30 ± 0.24
13	4	30	5	935.93 ± 0.52	263.51 ± 0.18
14	4	30	5	922.11 ± 0.32	257.45 ± 0.25
15	4	30	5	930.72 ± 0.45	260.90 ± 0.14
16	4	30	5	934.59 ± 0.91	265.09 ± 0.16
17	4	30	5	912.85 ± 1.56	258.09 ± 0.18

**Table 3 foods-13-03340-t003:** Analysis of variance for the regression model equation.

Source	Response Variables			
	TPC		GABA	
	F-Value	*p*-Value	F-Value	*p*-Value
Model	57.14	<0.0001 **	61.86	<0.0001 **
X_1_	11.03	0.0128 *	66.46	<0.0001 **
X_2_	18.54	0.0035 **	69.09	<0.0001 **
X_3_	63.42	<0.0001 **	0.15	0.7089
X_1_ X_2_	1.17	0.3143	0.95	0.3618
X_1_ X_3_	0.072	0.7962	6.79	0.0351 *
X_2_ X_3_	6.21	0.0414 *	0.33	0.5838
X_1_^2^	109	<0.0001 **	44.89	0.0003 **
X_2_^2^	128.25	<0.0001 **	215.47	<0.0001 **
X_3_^2^	133.09	<0.0001 **	113.87	<0.0001 **
Residual				
Lack of Fit	0.43	0.7415	1.01	0.4761
R^2^	0.9866		0.9876	
Adj R^2^	0.9693		0.9716	
Pred R^2^	0.9315		0.9034	

* Significant at *p* < 0.05; ** significant at *p* < 0.01.

**Table 4 foods-13-03340-t004:** Total phenol content (TPC), GABA, and individual phenolic contents of foxtail millet sprouts before and after optimization of the germination conditions.

	Phenolic Components	Individual Phenolic Content (mg/kg)	
		Pre-Optimized	Post-Optimized
free	*p*-hydroxybenzoic acid	7.20 ± 0.26 ^b^	46.92 ± 1.23 ^a^
	3-*p*-coumaroylquinic acid	6.57 ± 0.18 ^b^	40.89 ± 1.58 ^a^
	*p*-hydroxybenzaldehyde	1.00 ± 0.02 ^b^	8.86 ± 1.75 ^a^
	N-(*p*-coumaroyl) serotonin	8.27 ± 0.24 ^b^	42.46 ± 2.78 ^a^
	N-feruloylserotonin	8.77 ± 0.13 ^b^	44.82 ± 2.14 ^a^
	4-*p*-coumaroylquinic acid	14.37 ± 0.76 ^b^	47.62 ± 1.75 ^a^
	Feruloylquinic acid	19.97 ± 0.27 ^b^	62.02 ± 3.12 ^a^
	*trans*-ferulic acid	18.17 ± 0.36 ^b^	51.69 ± 2.15 ^a^
	Apigenin-C-pentosyl-C-hexoside	1.40 ± 0.10 ^b^	26.43 ± 1.75 ^a^
	1-O-*p*-coumaroyl-3-O-feruloylglycerol	6.47 ± 0.16 ^b^	38.31 ± 1.56 ^a^
	3,7-dimethylquercetin	37.97 ± 0.6 ^b^	45.43 ± 1.20 ^a^
bound	Protocatechuic acid	18.23 ± 0.9 ^b^	20.19 ± 1.42 ^a^
	Protocatechuic aldehyde	1.17 ± 0.02 ^b^	10.84 ± 0.89 ^a^
	*p*-hydroxybenzaldehyde	44.30 ± 1.0 ^b^	51.31 ± 3.85 ^a^
	Syringic acid	30.40 ± 0.48 ^b^	104.21 ± 5.85 ^a^
	*trans*-*p*-coumaric acid	1167.33 ± 6.8 ^b^	1386.38 ± 11.75 ^a^
	*cis*-*p*-coumaric acid	49.10 ± 0.15 ^a^	39.57 ± 1.75 ^b^
	*trans*-ferulic acid	2325.40 ± 1.78 ^b^	4032.13 ± 15.78 ^a^
	*cis*-ferulic acid	295.83 ± 3.79 ^a^	193.91 ± 2.85 ^b^
	TFA1	13.30 ± 0.14 ^b^	82.19 ± 2.41 ^a^
	8,5′-DFA	42.13 ± 0.2 ^b^	79.21 ± 1.75 ^a^
	8,8′-DFA	136.60 ± 1.78 ^b^	241.21 ± 9.72 ^a^
	TFA2	34.30 ± 0.73 ^b^	80.56 ± 3.12 ^a^
Free TPC (mg FAE/100 gDW)		157.31 ± 2.02 ^b^	402.72 ± 3.15 ^a^
Bound TPC (mg FAE/100 gDW)		344.02 ± 1.31 ^b^	523.80 ±2.65 ^a^
TPC (mg FAE/100 gDW)		501.33 ± 1.69 ^b^	926.53 ± 1.23 ^a^
GABA content (mg/kg)		159.73 ± 1.03 ^b^	259.13 ± 1.83 ^a^

Different letters in same row represent significant differences (*p* < 0.05).

## Data Availability

The original contributions presented in the study are included in the article, further inquiries can be directed to the corresponding author.

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
