# Peer review of "Optimization of Germination Conditions for Enriched γ-Aminobutyric Acid and Phenolic Compounds of Foxtail Millet Sprouts by Response Surface Methodology"

_foods, 2024, doi:10.3390/foods13203340_

Round 1

Reviewer 1 Report

Comments and Suggestions for Authors

Title:

It could be more specific, mentioning, for example, the critical compounds studied or the model used.

Abstract:

Further details of the results provide critical information to allow the reader to understand the magnitude of the findings and the relevance of the methods used without reading the entire article.

Introduction:

Expand the literature review, including previous studies on millet germination and similar techniques applied to other grains to optimize GABA and polyphenols. 

Given that foxtail millet is not as widely studied as other grains, it would be beneficial to delve deeper into its nutritional value and potential applications in the food industry. This would not only help to better justify the focus of the study but also underline the significance of your research in this less-explored area.

Materials:

It is important to include more detail in the specifications of the reagents, such as purity grade or suppliers, to ensure the reproducibility of the experiment by other investigators.

Methods:

Include more specific details on the process of polyphenol and GABA extraction, such as exact amounts of material used, extraction times, and precise temperatures. In addition, it would be advisable to explain whether calibrations of the instruments used were performed and how the accuracy of the measurements was ensured.

Experimental Design:

It would be helpful to justify why the Box-Behnken design was chosen instead of other statistical designs such as Plackett-Burman or full factorial designs. 

Statistical Analysis:

The assumptions of the ANOVA model, such as homogeneity of variances and normality, should be clearly described, and details should be provided on whether tests were performed to verify these assumptions. It would also be convenient to include confidence intervals for the effects of the treatments, improving the interpretation of the results.

It would be advisable to explicitly mention that the Design Expert software was used for the statistical analyses and, if applicable, indicate whether the necessary licenses are available.

Figures and Graphics:

The resolution and graphical representation of response surface and contour figures should be improved.

Comparison with other studies:

Comparing the results obtained with other similar studies in the literature helps contextualize the findings better and highlight the originality and relevance of the work within the field of study.

Statistical parameters of the model:

It is suggested to deepen the discussion of the statistical parameters of the model, such as PRESS and lack of fit, among others, to provide a more complete assessment of the validity of the fitted model.

Comparison of model and experimental results:

It would be useful to include the calculation of model accuracy or assertiveness in the comparison between model and experimental results. In addition, using tools such as those provided by the Design Expert software, it should be verified whether the predicted values are within the 95% confidence interval.

Discussion:

Under optimal germination conditions, a more in-depth discussion of the biochemical mechanisms underlying the increase in GABA and polyphenols should be included. A limitations section of the study would also be beneficial, demonstrating a critical and balanced reflection.

Conclusions:

Conclusions could be more concrete, providing quantitative values of critical results and focusing on the practical applications of these findings, which could attract the interest of both researchers and food industry professionals.

Comments on the Quality of English Language

Moderate editing of English language required.

Reviewer 2 Report

Comments and Suggestions for Authors

It is an interesting manuscript, optimizing the germination conditions for enriched GABA and phenolic compounds of foxtail millet sprouts. Here are some suggestions to improve the quality and presentation of the manuscript.

 Abstract: “The optimal germination conditions…….. were soaking at 31°C for 4.5 h and germinated with 4.5 g/L sucrose solution”. Which was the best time and temperature for germination??

In “..sprouts were 926.53 mgFAE/100 g” should indicate the value is per g of dm (mgFAE/100 gdm), according to the results.

“..with less difference from the predicted values.” Which was the predicted value?

Introduction

In “Setaria italica (L.) P. Beauv.” Only “Setaria italica” should be in italics.

Section 2.3.

The citation of methodology for free and bound phenolic extraction is missing. The author indicates “method previously reported” but does not add any citations.

In “The free and bound TPC were expressed as…. (mgFAE/100 g)” should indicate that the value es per gram of dry matter (mgFAE/100 gdm), according to the figure 1 and 2.

“The external method…….” Include the cite.

2.4. Extraction and GABA measurement.

In “as previously mentioned,” include the cite.

Section 2.5.1. Soaking conditions:

Replaces “…temperatures ranging from 15°C to 25 °C” with “temperatures ranging from 15 to 25°C”

Fig.1 and 2. What is “G” in X axis?? Please add the unit after the variable “soaking time (h),” soaking temperature (°C). Replace “2 h 4 h 6 h 8 h 10 h 12 h” with “2 4 6 8 10 12”; “15°C 20°C 25°C 30°C 35°C” with “15 20 25 30 35”.

Regarding the effect of temperature in soaking and germination conditions on GABA and TPC content, this analysis is not included in methodology.

Why not analyze the free and bound fractions independently?

Fig. 2. What is “S” in X axis?? The authors mentioned that TPC presents the highest level at 5 d. However, this value is non-statistical different from those obtained at 4 d.

Section 3.3.1. “The content of free phenolic compounds followed a similar trend with TPC, and the content of bound phenolic compounds was not significantly different at concentrations of 20 mM and 40 Mm of NaCl”. In this case, no statistical analysis is shown for free and bound fractions separately. This makes what has been discussed contradictory. The authors should mention that TPC represents the sum of free and bound phenolic fractions.

Fig. 3. Put the units “mM” and “g/L” after the variable in the X-axis and delete them in each treatment. Put the super index value where it is needed.

Check the nomenclature of treatments in “sucrose + CaCl2” and 6 mM CaCl2 + sucrose” because it seems that treatment “5 g/L sucrose + 6 mM CaCl2” is repeated several times, and the result is different. This also makes it difficult to understand and discuss the figure.

Section 3.3.4. Effects of the combination of sucrose and CaCl2 on GABA and TPC. This evaluation is not mentioned in the methodology section. Where is the treatment “5 g/L sucrose + 6 mM CaCl2 alone for germination” in Fig. 3.?

Table 2. Put the devest value or the letter from the media comparison test for both response variables.

Fig. 4. Add the unit after factor “soaking temperature”, “soaking time”, and “sucrose concentration”. Fig. 4d and 4h are the same as Fig. 4a and 4e. Do not repeat results. Delete these figures or present all interactions in contour plots. Also, for a better discussion, present the three-dimensional response surface plots for TPC and GABA content separately. The GABA content plots should be moved to section 3.4.2.

3.4.3. Experimental validation of the RSM mode

“…the conditions for germination of foxtail millet were modified to 4.5 h of soaking time, 31°C of soaking temperature, and 4.5 g/L of sucrose concentration”. Under these modified conditions, what was the predicted value??  What degree of data fit the model?

Replace “156 %, 52.23%, 84.41 %, and 6 %, respectively” with “156, 52.23, 84.41, and 6 %, respectively. Also, these increases are with respect to the pre-optimized conditions, but what was the increase compared to the control without germination? What was the real yield of all these parameters?

Section 3.5. Comparison of the individual phenolic levels

In “… Except for these two cis hydroxycinnamic acids, all the remaining bound phenolic derivatives showed a significant increase by optimizing germination conditions,”. To what is this increase attributed?

Please check the guide for authors and use the correct citation style. Put the references according to the guide for authors. You can use a recently published manuscript for a reference.

The authors cite about seven articles by the same authors (~18 % of the total references). Also, the manuscript has a similarity index of 28 %. Therefore, it is necessary to work on the wording and self-citations to reduce these values.

Round 2

Reviewer 1 Report

Comments and Suggestions for Authors

Accept.

Reviewer 2 Report

Comments and Suggestions for Authors

I have revised the paper, and the authors have addressed the comments, and have reduced the level of similarity and self-citation, improving the quality of the manuscript.